5

# **Quantifying Resilience in Non-Autonomous and Stochastic Earth System Dynamics with Application to Glacial-Interglacial Cycles**

Jakob Harteg<sup>1,2</sup>, Nico Wunderling<sup>1,3,4</sup>, and Jonathan F. Donges<sup>1,5,6</sup>

Correspondence: Jakob Harteg (jakob.harteg@pik-potsdam.de) and Jonathan F. Donges (jonathan.donges@pik-potsdam.de)

**Abstract.** Understanding Earth resilience—the capacity of the Earth system to absorb and regenerate from perturbations—is key to assessing risks from anthropogenic pressures and sustaining a safe operating space for humanity within planetary boundaries. Classical resilience indicators are designed for autonomous systems with fixed attractors, but the Earth system is fundamentally non-autonomous and out of equilibrium, calling for new ways of defining and quantifying resilience.

Here, we introduce a path-resilience approach that assesses how perturbations deviate from and return to a reference trajectory of a conceptual climate model replicating the glacial-interglacial cycles of the Late Pleistocene. We generate two types of perturbation ensembles: a stochastic ensemble and a single-event ensemble, and compute two complementary metrics: the Reference Adherence Ratio (RAR)—defined as the fraction of stochastic trajectories that remain within a narrow band around the unperturbed trajectory—and return time—defined as the time a single perturbed trajectory takes to return to the reference path. Together, these metrics reveal strong temporal variation in resilience across the glacial-interglacial cycles. We find that RAR increases markedly during deglaciations and peaks in interglacial periods, while return times generally shorten as the system approaches interglacial conditions—indicating that certain phases of the cycles act as convergence zones and potential anchors of Earth system stability. As the Earth system departs from such stable interglacial regimes under ongoing anthropogenic forcing, understanding the resilience of these trajectories—and what it may take to return to them—becomes increasingly important.

These results highlight that resilience in non-autonomous systems is inherently path-dependent and illustrate a promising first step toward its quantification. Further research is needed to develop more general resilience indicators suitable for complex, forced dynamical systems.

<sup>&</sup>lt;sup>1</sup>Potsdam Institute for Climate Impact Research, Member of the Leibniz Association, Telegrafenberg A 31, 14473 Potsdam, Germany

<sup>&</sup>lt;sup>2</sup>Institute for Physics and Astronomy, University of Potsdam, Karl-Liebknecht-Str. 24/25, 14476 Potsdam-Golm, Germany <sup>3</sup>Center for Critical Computational Studies, Goethe University Frankfurt, Theodor-W.-Adorno-Platz 1, 60323 Frankfurt am Main, Germany

<sup>&</sup>lt;sup>4</sup>Senckenberg Research Institute and Natural History Museum, Senckenberganlage 25, 60325 Frankfurt am Main, Germany <sup>5</sup>Department Integrative Earth System Science, Max Planck Institute of Geoanthropology, Kahlaische Strasse 10, 07745 Jena, Germany

<sup>&</sup>lt;sup>6</sup>Stockholm Resilience Centre, Stockholm University, Albanovägen 28, SE-106 91 Stockholm, Sweden

40

50

#### 1 Introduction

Quantifying Earth system resilience is essential for evaluating the planet's capacity to withstand and regenerate from growing human-induced stresses, such as greenhouse gas emissions and land-use changes (Rockström et al., 2021; Anderies et al., 2023). As humanity continues to transform the climate system and the biosphere, understanding the processes that strengthen or undermine its stability becomes increasingly important (Yi et al., 2024). Such knowledge is crucial for developing strategies to regenerate and revitalise Earth system resilience, ultimately helping society return to a safe operating space within planetary boundaries (Richardson et al., 2023; Caesar et al., 2024; Folke et al., 2010).

The concept of resilience has been explored from multiple disciplinary perspectives, giving rise to diverse definitions. Two predominant forms have traditionally defined the field: engineering resilience (Holling, 1996), which concerns linear stability analysis and the rate of return to equilibrium after small perturbations, and ecological resilience, which addresses the resilience of nonlinear dynamical systems with multiple stable states and their capacity to maintain essential functions while transitioning between different regimes (Holling, 1973). Related terms such as stability, robustness, and persistence are often used interchangeably despite subtle differences, creating definitional diversity that has spread across social and natural sciences (Krakovská et al., 2024; Anderies et al., 2013). To navigate this complexity, the question of *resilience of what to what?* (Holling, 1973; Tamberg et al., 2022) has become a guiding framework — clarifying which system components are meant to be sustained, which properties are at risk, and which disturbances are relevant. Here, this question is interpreted as the resilience of the glacial-interglacial ice volume trajectory to perturbations, i.e., the capacity of the ice-climate-carbon system to remain close to its characteristic large-scale path in face of external shocks.

This framework gains particular relevance when examining Earth's climate history. Palaeoclimate variability during the Cenozoic offers important insights into the Earth system's changing stability and resilience, as there is evidence for alternative attractors (Westerhold et al., 2020) and resilience loss before major climate shifts and hyperthermal events in that period (Setty et al., 2023).

Building on this palaeoclimate context, the glacial-interglacial cycles of the Late Pleistocene offer a compelling test case for studying Earth system resilience, due to their regularity and sensitivity to external forcing (Milankovitch, 1920; Lisiecki and Raymo, 2005). These cycles involve transitions between long glacial periods with large continental ice sheets and shorter interglacials with reduced ice volume, paced by variations in Earth's orbital configuration—eccentricity, obliquity, and precession—which act faster than the system's internal adjustment times. The dominance of the  $\sim 100,000$ -year cycle in the Late Pleistocene, despite weak direct insolation forcing at that frequency, poses a long-standing puzzle in palaeoclimate dynamics (Imbrie et al., 1993), with a variety of proposed solutions (e.g., Willeit et al., 2019; Abe-Ouchi et al., 2013). The Earth system can thus be viewed as a non-equilibrium, externally forced system, following a quasi-periodic trajectory shaped by the interplay between astronomical forcing and internal feedbacks.

Over the past 800,000 years, this glacial-interglacial rhythm has been remarkably persistent (Lisiecki and Raymo, 2005). However, model studies suggest that anthropogenic greenhouse gases already present in the atmosphere may delay the next glacial inception by over 100,000 years (Ganopolski et al., 2016), marking a significant departure from the quasi-periodic limit

cycle that has dominated recent Earth history. Continued warming raises the risk of a further shift toward "Hothouse Earth" conditions that last occurred millions of years ago, conditions that modern human societies would likely be unable to adapt to (Steffen et al., 2018; Kaufhold et al., 2025). The resilience of Holocene-like interglacial states, which define the safe operating space delineated in the planetary boundary framework (Rockström et al., 2024), must therefore be understood within a non-equilibrium framework, as such interglacial states naturally occur as part of a dynamically evolving glacial-interglacial cycle. Observational evidence already hints at systemic weakening, including reduced carbon sink capacity in land-systems (Ke et al., 2024) particularly in the Amazon (Brienen et al., 2015; Gatti et al., 2021), and amplified feedbacks in the ocean-atmosphere system and other tipping elements (Wunderling et al., 2024). While this study does not address such future scenarios directly, they underscore the need to understand Earth system resilience.

Classical resilience indicators—such as return time to equilibrium, eigenvalue-based stability measures, or potential well depth—are poorly suited for systems like the glacial-interglacial cycle. These metrics are typically developed for autonomous systems with fixed-point attractors and assess resilience as the rate of return to such a state after perturbation (Krakovská et al., 2024). The Earth system, by contrast, is fundamentally non-autonomous, driven by time-dependent forcings such as orbital variations and, more recently, anthropogenic influences. These drivers operate on shorter timescales than the system's intrinsic adjustment times, resulting in transient dynamics that do not converge to equilibrium. Consequently, conventional indicators fail to meaningfully characterise resilience in this context.

To address this limitation, a conceptual shift is made toward *path resilience* (Anderies et al., 2023). Rather than asking whether the system returns to a stable state, the focus here is on how perturbed trajectories deviate from — and eventually return to — a reference trajectory. In this framing, the path itself becomes the object of resilience analysis, making it more appropriate for externally forced, non-equilibrium systems. Returning to the "resilience of what to what?" framework, resilience is defined here as the resilience of the glacial-interglacial ice volume trajectory to perturbations in ice volume, with the "sustainant" being the historical reference trajectory, and the "disturbance" being imposed stochastic variability or pulse-like shocks, representing internal variability and external shocks respectively.

In this study, we employ a reduced-complexity climate model developed by Talento and Ganopolski (2021). The model captures essential large-scale feedbacks among global ice volume, atmospheric CO<sub>2</sub>, and global mean temperature. Its low dimensionality makes it computationally efficient and particularly suitable for long-term ensemble simulations designed to explore dynamical behaviour under perturbations. Our goal is not to produce detailed predictions, but to develop and demonstrate a transparent methodology for measuring path-wise Earth system resilience that can be extended to more complex models.

We generate two perturbation ensembles: a stochastic ensemble with low-amplitude continuous noise, following the Hasselmann framework (Hasselmann, 1976), to represent unresolved fast processes; and a single-event ensemble, in which the system is shocked once at a selected time point and then allowed to evolve deterministically. We apply two complementary diagnostics: the Reference Adherence Ratio (RAR), which measures how closely perturbed trajectories follow a reference path; and the return time, which quantifies how long it takes trajectories to rejoin and remain near the reference. Together, these metrics operationalise path resilience in a non-autonomous dynamical systems setting.

The paper is organised as follows: Section 2 describes the model setup and governing equations; Section 3 introduces the perturbation ensembles and resilience metrics, and presents the results; Section 4 discusses model limitations and their implications; and Section 5 summarises the main conclusions and outlines future directions.

# 90 2 Model Setup

100

This section describes the reduced-complexity climate model of Talento and Ganopolski (2021) used to simulate glacial-interglacial dynamics on orbital timescales. The model represents the co-evolution of three globally averaged variables: ice volume anomaly v(t), atmospheric  $CO_2$  concentration  $CO_2(t)$ , and global mean temperature anomaly T(t). These components are coupled through a single first-order differential equation describing ice sheet evolution, alongside two diagnostic relationships for temperature and atmospheric carbon dioxide.

The model is explicitly non-autonomous, with time-dependent orbital forcing as the sole external driver. Internal feedbacks include the ice-albedo effect, dynamic carbon cycle responses, and a memory term reflecting the persistence of ice sheets over multi-millennial timescales. These mechanisms jointly enable the model to reproduce key features of Quaternary glacial variability, such as the characteristic asymmetric ice volume cycles.

The model is governed by three coupled equations. The global ice volume evolution is described by:

$$\frac{dv}{dt} = \frac{b_1 v - b_2 v^{3/2} - b_3 (f - \bar{f}) - b_4 \log(\text{CO}_2)}{1 - b_5 M_v} + b_6,\tag{1}$$

where f(t) is the time-dependent orbital forcing,  $\bar{f}$  its long-term mean,  $\log$  denotes the natural logarithm, and  $M_v(t)$  a memory term that accounts for the integrated ice volume over the past  $\tau = 30$  kyr:

$$M_v(t) = \begin{cases} \frac{1}{\tau} \int_{t-\tau}^t v(x) \, dx & \text{if } \frac{dv}{dt} 

120

Finally, global mean temperature anomaly (in units of K) is given by:

$$T = d_1 v + d_2 \log \left(\frac{\text{CO}_2}{278}\right),\tag{4}$$

where the first term represents the ice-albedo effect, and the second captures the greenhouse effect of CO<sub>2</sub> approximated by its logarithmic dependence on concentration relative to the pre-industrial baseline.

Figure 1 compares the simulated ice volume trajectory with the palaeo-reconstruction from Spratt and Lisiecki (2016), along with the imposed orbital forcing and frequency spectra. Using a model calibration as described in Section 2.2, the model captures both the amplitude and dominant periodicities of Quaternary glacial cycles reasonably well, making it a solid basis for the perturbation experiments in this study.

Figure 1. Model validation against palaeo-reconstructions over the past 800 kyr. (a) orbital forcing anomaly f' = f - mean(f). (b) ice volume anomaly from the model compared to palaeo reconstructions from Spratt and Lisiecki (2016), with Marine Isotope Stage (MIS) interglacial boundaries marked. (c) frequency spectra of the model output compared to the palaeo reconstruction.

In the model, orbital forcing f represents maximum summer insolation at 65°N, which drives the glacial-interglacial cycles (Laskar et al., 2004). Talento and Ganopolski used a parameterisation in which insolation was decomposed into obliquity obl(t) and climatic precession pre(t) components, using a least-squares fit as in Jackson and Broccoli (2003), and recombined into a scaled linear combination to form a proxy for orbital forcing:

125 
$$f(t) = \operatorname{pre}(t) + \gamma(\operatorname{obl}(t) - \langle \operatorname{obl} \rangle),$$
 (5)

with  $\gamma=1.04$  optimised to maximise the correlation with the critical  ${\rm CO_2}$  threshold. The mean orbital forcing  $\bar{f}$  entering the ice volume equation is calculated over a [-1000, +1000] kyr window to define deviations from long-term average insolation. The orbital forcing dataset is available online (Talento, 2021) and is applied here without modification.

## 2.1 Model Constraints and Limitations

Like all reduced-complexity models, this model employs simplifying assumptions and explicit constraints. These choices are detailed in the original study by Talento and Ganopolski (2021), but we briefly highlight two constraints that are important for this work.

The most important constraint for our resilience analysis is that global ice volume is constrained not to fall below its preindustrial level ( $v \ge 0$ ), reflecting the present-day as the zero anomaly baseline. This approach leads to interglacials being represented by extended periods of zero ice-volume anomaly, which does not fully capture the gradual transitions seen in palaeoclimate data. This artificial boundary may create an illusion of enhanced stability during interglacial periods, as perturbed trajectories cannot deviate below this constraint.

The model also imposes a constraint to simulate the mid-Brunhes transition, a shift around 400 kyr BP after which interglacials became notably stronger. This change is not well explained by orbital forcing alone and is likely linked to external factors not captured by the model. To approximate this behaviour, ice volume is not allowed to fall below 0.05 before 400 kyr BP. This, however, introduces a sharp drop from 0.05 to 0 at the time of the transition, which does not realistically reflect the gradual nature of the transition.

# 2.2 Numerical Implementation

150

Talento and Ganopolski (2021) originally implemented the model in MATLAB. We re-implemented it in Python and integrated it using a forward Euler scheme with a time step of  $\Delta t=1$  kyr, consistent with the original study. The memory term  $M_v(t)$  is calculated using a trapezoidal approximation of the integral over the past 30 kyr. This numerical setup provides sufficient accuracy to reproduce key features of glacial-interglacial cycles while remaining computationally efficient for ensemble experiments. To ensure the model reaches a steady state before comparison with palaeo data, each simulation is extended by 200 kyr prior to the start of the observational window.

To calibrate the model's nine tuneable parameters, we employed a Metropolis-Hastings Markov Chain Monte Carlo (MCMC) approach. We follow the original calibration strategy and optimise the model ice volume output by comparing it against reconstructed ice volume anomalies from the Spratt and Lisiecki (2016) sea-level stack spanning the past 800 kyr, and obtain a Pearson correlation coefficient of 0.892 with the reconstruction. Notably, the parameters  $c_2$  and  $c_3$ , which govern the sensitivity of  $CO_2$  to ice volume and its rate of change (Eq. 3), differ substantially from the original values. This may reflect structural dif-

170

ferences in implementation or sensitivity in the calibration process. Further details on the MCMC setup and likelihood function are provided in Appendix A.

#### 3 Perturbation Experiments and Indicators of Path Resilience

#### 3.1 Experiment I: Stochastic Ensemble and Reference Adherence Ratio (RAR)

To investigate the resilience of the glacial-interglacial trajectory under persistent, small-scale disturbances (e.g. representing internal variability), we extend the model with a stochastic noise component and analyse its effect on an ensemble of simulations. Specifically, we generate 20,000 ice volume trajectories subject to continuous Gaussian perturbations with standard deviation  $\sigma = 0.001$ . This ensemble allows us to assess how closely the system remains aligned with its unperturbed reference path over time. To quantify this behaviour, we introduce the *Reference Adherence Ratio* (RAR), a non-autonomous resilience indicator that measures the fraction of ensemble members staying within a tolerance band of  $\alpha = 0.001$  around the reference trajectory. In the following, we first describe the stochastic extension to the model, then introduce the RAR and present the results.

## 3.1.1 Stochastic Extension to the Model

We extend the model with a stochastic component following Hasselmann's approach to stochastic climate modelling (Hasselmann, 1976), where unresolved processes are modelled as noise. A general Stochastic Differential Equation (SDE) can be written as

$$dX_t = \mu(X_t, t) dt + \sigma(X_t, t) dW_t, \tag{6}$$

where  $X_t$  is the state variable at time t,  $\mu$  is the deterministic drift term,  $\sigma$  is the diffusion term controlling the strength of random fluctuations, and  $dW_t$  is a Wiener process (Brownian motion) with independent Gaussian fluctuations of mean zero and variance proportional to the time step.

We apply this to the ice volume equation, which is the only differential equation in our model. Millennial-scale oscillatory variability in the polar cryosphere is well documented, including Dansgaard-Oeschger events (Dansgaard et al., 1989) and Heinrich events (Heinrich, 1988), which may be driven by internal ice sheet mechanisms, shifts in ocean heat transport, and changes in atmospheric circulation or solar variability (Berger et al., 2016; Ghil and Lucarini, 2020).

Taking  $\mu = dv/dt$ ,  $X_t = v(t)$  and  $\sigma = {\rm const}$ , we obtain the following stochastic differential equation for ice volume:

$$dv = \left(\frac{b_1 v - b_2 v^{3/2} - b_3 (f - \bar{f}) - b_4 \log(\text{CO}_2)}{1 - b_5 M_v} + b_6\right) dt + \sigma dW_t.$$
 (7)

We compute  $dW_t$  by sampling from a normal distribution with mean zero and standard deviation  $\sqrt{\Delta t} = 1$  (since  $\Delta t = 1$  kyr, the time step of the simulation), formally replacing the Explicit Euler Method with the Euler-Maruyama method (Maruyama,

1959). This enables us to run model ensembles with different noise realisations and investigate alternative ice volume trajectories and their statistics.

#### 3.1.2 Noise Amplitude Selection

The amplitude of stochastic forcing,  $\sigma$ , is a key parameter controlling the strength of internal variability represented in the model. To find an appropriate value, we first consider the standard deviation of residuals between modeled and observed ice volume trajectories ( $\sigma \approx 0.1$ ) as a physically motivated starting point. However, this value causes the characteristic 100 kyr spectral peak to disappear, see Figure 2. Reducing the noise to  $\sigma = 0.001$  restores the 100 kyr peak while maximising ensemble spread within the constraints of spectral consistency. Figure 2 explores the impact of varying  $\sigma$  on model behavior and spectral properties. As noise increases, trajectories become more dispersed and the glacial-interglacial cycles are obscured. Even at lower noise levels, substantial deviations from the deterministic reference can occur, particularly during glacial periods. We use  $\sigma = 0.001$  for subsequent resilience analyses, as it preserves the key spectral features of the glacial-interglacial cycle.

Figure 2. Impact of varying noise levels on stochastic ice volume trajectories and their frequency spectra. (a) 500 stochastic ice volume trajectories for different noise levels  $\sigma \in \{0.1, 0.01, 0.001, 0.0001, 0.0001\}$ . (b) associated mean frequency spectra (black line) compared to palaeo data Spratt and Lisiecki (2016) (grey shading).

## 3.1.3 Reference Adherence Ratio (RAR)

To quantify how strongly the ensemble trajectories cluster around the reference path, we introduce the Reference Adherence Ratio (RAR). This metric measures the fraction of ensemble members that remain within a tolerance band of width  $\alpha$  around the reference trajectory—the deterministic trajectory—at each time step:

$$RAR(t) = \frac{1}{N} \sum_{i=1}^{N} \mathbf{1} (|v_i(t) - v_{ref}(t)| 

Figure 3. Reference Adherence Ratio (RAR) as a measure of Earth system resilience. (a) 20,000 stochastic ice volume trajectories (black) and the reference (deterministic) trajectory (pink). (b) RAR over time, quantifying the proportion of ensemble members that remain within a small tolerance ( $\alpha = 0.001$ ) of the reference. Interglacial periods (highlighted in pink) are associated with markedly higher RAR values, indicating enhanced resilience.

# 3.2 Experiment II: Single-Event Perturbation Ensemble and Return Time

To complement the stochastic ensemble, we also assess the system's resilience to isolated, abrupt perturbations using a singleevent perturbation ensemble. This setup probes the system's ability to recover from discrete shocks applied at specific times along the reference trajectory, rather than from continuous background variability.

At each time step within the simulation window  $t \in [-800, -1]$  kyr, we generate an ensemble of 20 trajectories by applying a one-time perturbation to the ice volume variable, with an amplitude drawn uniformly from the interval [-0.01, +0.01], resulting in a total of 16,000 perturbed trajectories. The system then evolves deterministically from that point onward. This perturbation amplitude is an order of magnitude larger than the stochastic noise used in the previous experiment ( $\sigma = 0.001$ ), and is meant to represent more shock-like disturbances.

## 225 3.2.1 Definition of Return Time

We define the *return time*  $T_R$  as the first time a perturbed trajectory re-enters a narrow band around the reference trajectory and stays within it for the remainder of the simulation:

$$T_R = \inf\left\{t \ge 0 : \forall \tau \ge t, |\tilde{v}(\tau) - v(\tau)| 

where  $\tilde{v}$  is the perturbed trajectory, v is the reference trajectory, and  $\alpha$  is a strict convergence threshold. We run each trajectory 1000 kyr beyond the perturbation to allow sufficient time for return. If the perturbation does not push the trajectory outside the  $\alpha$ -band initially, we assign  $T_R = 0$ .

Return time provides a temporal measure of path resilience, indicating how quickly the system recovers from an isolated shock. Short return times suggest high resilience; long return times indicate prolonged deviation. Conceptually, this aligns with classical engineering resilience metrics (Krakovská et al., 2024) but is applied here to recovery toward a reference *trajectory*, not a fixed point.

Our return criterion is deliberately strict: unlike exponential decay-based definitions (e.g., 1/e recovery), it requires sustained convergence. This avoids falsely classifying temporary re-approaches as recovery—particularly important in a non-autonomous system where external forcing may later drive re-divergence.

# 3.2.2 Return Time Results and Interpretation

We compute return times using a strict convergence threshold of  $\alpha = 10^{-6}$ , chosen to ensure persistent re-alignment of perturbed and reference trajectories. Figure 4 presents the results: Figure 4a shows 16,000 perturbed trajectories (black) alongside the unperturbed reference (pink), while Figure 4b displays the distribution of return times for perturbations applied at each time step. The black line shows the median return time, and the blue shaded region marks the full range across all perturbations.

Return times vary substantially over the glacial-interglacial cycle, generally decreasing as the system approaches interglacial conditions. In particular, MIS11 and the Holocene stand out as strong convergence zones, with many long excursions returning to the reference path. This suggests higher resilience during these periods. One likely explanation is their unusual length: both MIS11 and the Holocene are exceptionally long interglacials due to orbital forcing, providing extended windows for perturbed trajectories to realign (Past Interglacials Working Group of PAGES, 2016).

Sensitivity tests using larger perturbation amplitudes and more ensemble members (up to 100 per time step) increased the spread of return times but did not change the overall temporal pattern. Likewise, moderately relaxing the convergence threshold  $\alpha$  shifts absolute return times but preserves the key structure of the results. This confirms that our findings are robust to these parameter choices.

Figure 4. Return time to reference trajectory as a measure of Earth system resilience. (a) Ice volume trajectories from 16,000 single-event perturbations (black) and the unperturbed reference trajectory (pink) over the last 800 kyr. Interglacial periods are highlighted in pink. (b) Distribution of return times for perturbations applied at each time step. The black line shows the median return time, while the blue shaded region indicates the full range (minimum to maximum) of return times across all perturbations at each time. Shorter return times correspond to higher resilience. Return time is defined relative to a strict convergence threshold of  $\alpha = 10^{-6}$ .

## 4 Discussion

In this study, we introduced a perturbation-based approach to exploring Earth System resilience, aiming to move beyond traditional attractor-based concepts. By analysing how perturbed trajectories deviate from and return to a reference path, we quantified the system's time-varying resilience over the last 800 kyr of glacial-interglacial dynamics. Our results reveal a strongly state-dependent pattern: during some periods, perturbed trajectories remain tightly clustered around the reference, while during others they diverge widely.

Our stochastic ensemble results align with previous findings by Mitsui and Crucifix (2016), who showed that the timing of stochastic forcing in conceptual climate models can influence the duration and nature of transient excursions. We build on this by explicitly quantifying such detours using resilience metrics—specifically the Reference Adherence Ratio (RAR) and return time—and showing that sensitivity to perturbations depends systematically on the system's position along the glacial cycle.

In particular, MIS11 and the Holocene stand out as strong convergence points where many trajectories recover. Their unusual length and stability, a feature also emphasised in model-based studies of glacial pacing dynamics (Ganopolski, 2024), may explain this exceptional behaviour. More generally, interglacial periods consistently show lower variance in the ensemble, with perturbed trajectories tending to cluster more tightly around the reference path compared to glacial periods, which aligns with paleoclimate evidence showing that natural temperature variability was lower during interglacial conditions (Rehfeld et al., 2018). These findings suggest that Earth System resilience not only differs between glacial and interglacial states, with interglacials being more resilient, but also peaks during key transition periods when the system rapidly reorganises. Some of

these patterns may, however, be influenced by structural features of the model. We address these limitations in more detail below.

#### 4.1 Model Limitations

Several limitations of the model should be kept in mind when interpreting the results. Most importantly, the apparent resilience during interglacials may partly be an artefact of a hard lower bound imposed on ice volume ( $v \ge 0$ ), which prevents trajectories from falling below pre-industrial levels. This constraint leads to extended periods of zero ice volume and artificially suppresses variability during interglacials, potentially inflating both RAR and return time convergence. Future work should compare results across models without such constraints to assess the extent of this effect.

A further limitation lies in the model's non-Markovian structure. It includes a memory term that integrates ice volume over the past 30 kyr, making it effectively infinite-dimensional and preventing the use of standard Jacobian-based stability analysis. This justifies our simulation-based approach, but also limits direct comparisons with studies that rely on linearised stability frameworks.

Finally, the model lacks self-sustained oscillations, which have been proposed as key ingredients in explaining the 100,000-year glacial pacing problem (Ganopolski, 2024). Our results therefore reflect the resilience of a purely forced system and may change in models that include internally driven oscillations. A natural next step is to validate the findings with a more complex model, but given the need for large ensembles, there are computational challenges. It may be possible using Earth system models of intermediate complexity like CLIMBER-X, which can simulate 10,000 simulation years per day on a node with 16 CPUs (Willeit et al., 2022, 2023). If run on a more powerful cluster, one could generate a single 1 million year simulation in around 20 days, but it becomes clear that large ensembles are limited. Fully integrated Earth system models would require computational resources that make comparable ensemble experiments currently impractical.

Despite these limitations, our approach connects to a broader shift in how resilience is conceptualised and operationalised in non-autonomous systems, as discussed next.

# 4.2 Relationship to Existing Research

Classical resilience indicators are designed for autonomous systems near equilibrium, where concepts like basins of attraction and return time to fixed points are well-defined. Several studies have highlighted the lack of tools for analysing resilience in forced, transient systems (Krakovská et al., 2024). Our approach addresses this gap by using ensemble perturbations to evaluate system stability over time, offering a first step toward quantifying resilience in non-autonomous systems.

We see this work as a practical example of path resilience thinking, applied to palaeoclimate dynamics. The concept of path resilience (Anderies et al., 2023) shifts the focus from return to equilibrium states to return to a reference trajectory—making it more appropriate for externally forced systems like the glacial-interglacial cycle.

Related work by Medeiros et al. (2023) uses time-varying Jacobians and eigenvectors to assess sensitivity in ecological communities. While powerful, their method assumes differentiable, Markovian dynamics. In contrast, our model includes a

memory term—an integral over past ice volume—which makes standard Jacobian-based methods inapplicable. This motivates our use of a model-agnostic, simulation-based strategy that can handle non-Markovian dynamics.

Our approach builds on Hasselmann's programme for stochastic climate modelling (Hasselmann, 1976), where unresolved processes are treated as noise. This perspective laid the foundation for analysing climate variability through probabilistic ensemble dynamics and has recently been extended by Lucarini and Chekroun (2023) and others through operator-theoretic methods. In particular, transfer operator theory provides a systematic framework for describing the evolution of probability distributions in phase space and has been used to study mixing, regime transitions, and responses in high-dimensional systems. These methods offer conceptual and computational tools that could support more general resilience indicators—especially in systems with complex dynamics and non-equilibrium behaviour.

However, while transfer operator methods are well developed for autonomous, Markovian systems, they are not yet fully applicable to systems with memory (non-Markovian dynamics), and a general theoretical framework for analysing stability and resilience in non-autonomous systems—such as those driven by orbital forcing—remains underdeveloped (Froyland et al., 2010; Lucarini and Chekroun, 2023).

#### 315 5 Conclusion

As humanity continues to push the Earth system toward novel and potentially unstable states, understanding the resilience of both past and projected trajectories becomes increasingly critical. The approach developed here represents a step toward that goal by enabling resilience analysis in systems that do not settle into traditional equilibrium states.

We introduced a simulation-based method to assess Earth system resilience across glacial-interglacial cycles. By applying perturbations to a reduced-complexity climate model and analysing trajectory responses, we show that resilience varies markedly over time and depends on the system's state at the moment of disturbance.

Our findings reveal that resilience fluctuates throughout the glacial-interglacial cycle, reflecting the evolving dynamical landscape of the system. Interglacial periods—particularly MIS11 and the Holocene—consistently draw perturbed trajectories back toward the reference path, suggesting enhanced recovery capacity linked to their unusual duration. Deglaciation phases show similar behaviour, with trajectories reconverging even before interglacial conditions are reached. The reference adherence ratio and return time provide useful, albeit preliminary, indicators of resilience in non-autonomous systems.

These results mark initial progress toward quantifying resilience in settings where traditional equilibrium-based tools fall short. The path-resilience framework shifts focus from fixed-point attractors to the stability of entire trajectories—an essential perspective for assessing the resilience of transient climate states as the Earth system moves into uncharted territory under anthropogenic forcing.

Several limitations must be acknowledged. The model's hard lower bound on ice volume may inflate apparent resilience during interglacials, and its non-Markovian structure prevents standard stability analysis. Moreover, the model lacks self-sustained oscillations that could influence glacial pacing. Despite these simplifications, the study demonstrates the potential of trajectory-based resilience assessment in externally forced systems.

Future work should test and extend this framework across other conceptual and intermediate-complexity models such as CLIMBER-X (Willeit et al., 2022, 2023) and LOVECLIM (Goosse et al., 2010). Developing more advanced resilience indicators for non-autonomous dynamics remains a key priority—promising directions include transfer operator theory, pullback attractors, random dynamical systems, and finite-time Lyapunov exponents.

This line of research aligns with Hasselmann's view of the climate as a stochastically forced dynamical system. Recent work by Lucarini and colleagues (Lucarini and Chekroun, 2023) has begun to formalise this perspective, paving the way for robust, operator-based resilience metrics. Advancing these ideas could deepen our understanding of Earth system resilience and support the development of general tools for resilience assessment.

Code availability. Code and data used in this study are available at https://doi.org/10.5281/zenodo.16603222 (Harteg, 2025).

## Appendix A: Model Calibration via Metropolis-Hastings MCMC

To recalibrate the model described by Talento and Ganopolski (2021), a Metropolis-Hastings Markov Chain Monte Carlo (MCMC) algorithm (Hastings, 1970) was implemented, targeting agreement between simulated and reconstructed ice volume anomalies over the past 800 kyr based on the sea-level stack from Spratt and Lisiecki (2016). The recalibration achieved a Pearson correlation coefficient of 0.892.

The algorithm proceeds as follows:

- 1. An initial model run m(p) is generated using the published parameter set p from Talento and Ganopolski (2021), as shown in Table A1.
  - 2. One parameter  $p_i$  is randomly selected and perturbed by a small factor drawn from a normal distribution  $\mathcal{N}(1,\sigma)$  with  $\sigma = 0.05$ :

$$\hat{p}_i = \mathcal{N}(1, 0.05) \cdot p_i, \tag{A1}$$

resulting in a new candidate parameter set  $\hat{p}$ .

3. A model run  $\hat{m}(\hat{p})$  is performed using the perturbed parameters, and the likelihood of the resulting ice volume time series is computed as:

$$L = -\sum_{t=1}^{t_{\text{end}}} [v_{\text{model}}(t) - v_{\text{palaeo}}(t)]^2 - P_{\text{LGM}} - P_{\text{CO}_2},$$
(A2)

where  $P_{\rm LGM}$  penalises poor agreement with the Last Glacial Maximum:

$$P_{\text{LGM}} = \alpha \left[1 - v(t_{\text{LGM}})\right]^2 \quad \text{with} \quad \alpha = 1000, \tag{A3}$$

and  $P_{\rm CO_2}$  discourages solutions with unrealistically high  ${\rm CO_2}$  concentrations:

$$P_{\text{CO}_2} = \sum_{t=1}^{t_{\text{end}}} \left( \max\left[0, \text{CO}_2(t) - 300\right] \right)^2.$$
(A4)

4. The perturbed model is accepted with probability:

$$p_{\text{accept}} = \min\left(1, \exp(\hat{L} - L)\right),\tag{A5}$$

where  $\hat{L}$  is the likelihood of the candidate model. Otherwise, the previous model is retained.

5. Steps 2-4 are repeated for N = 5000 iterations per chain.

To ensure robustness, the procedure was repeated for 1000 independent chains, each with a different random seed. The best-fitting model across all runs achieved a maximum correlation of 0.892, with the corresponding parameter values listed in Table A1. These calibrated parameters were used for all simulations in this study unless stated otherwise.

**Table A1.** Unrounded model parameters from Talento and Ganopolski (2021) and from the Metropolis-Hastings MCMC recalibration performed in this study. These parameters were used for all simulations unless stated otherwise.

| Parameter        | Talento and Ganopolski (2021) | This study (MCMC)     |
|------------------|-------------------------------|-----------------------|
| $\overline{b_1}$ | 0.22                          | 0.21225019974916653   |
| $b_2$            | 0.29                          | 0.2836925345028339    |
| $b_3$            | 0.0008                        | 0.0008174423131295259 |
| $b_4$            | 0.095                         | 0.095                 |
| $b_5$            | 0.18                          | 0.18964641973723284   |
| $b_6$            | 0.53                          | 0.5261626610099003    |
| $c_1$            | 17.28                         | 17.824476083652947    |
| $c_2$            | -31.95                        | -24.49497200960442    |
| $c_3$            | -120.0                        | -47.76302777426819    |
| $c_4$ (fixed)    | 278                           | 278                   |
| $d_1$ (fixed)    | -3                            | -3                    |
| $d_2$ (fixed)    | 5.56                          | 5.56                  |

# 370 Appendix B: RAR Sensitivity to Tolerance

Figure B1 shows the Reference Adherence Ratio (RAR) computed across 50 different absolute tolerance values ranging from  $5 \times 10^{-5}$  to  $5 \times 10^{-3}$ . While the absolute magnitude of RAR varies with the threshold, the timing and structure of peaks and troughs remain consistent, indicating that RAR captures robust features of the system's path resilience across a wide parameter range.

Figure B1. Sensitivity of the Reference Adherence Ratio (RAR) to different convergence thresholds,  $\alpha$ . (a) unperturbed reference trajectory (black) and 1,000 stochastic ensemble members (pink). (b) RAR values computed for 50 tolerance levels (colour scale), ranging from  $5 \times 10^{-5}$  to  $5 \times 10^{-3}$ . Results are computed for 1,000 trajectories.

Author contributions. JH, NW, and JFD designed the research. JH implemented the model, performed the analysis, and wrote the manuscript. All authors reviewed the manuscript, contributed to the discussion and interpretation of the results.

Competing interests. JFD is a member of the editorial board of Earth System Dynamics. The authors declared no further competing interests.

Acknowledgements. This is ClimTip contribution #82; the ClimTip project has received funding from the European Union's Horizon Europe research and innovation programme under grant agreement No. 101137601: Funded by the European Union. Views and opinions expressed are however those of the author(s) only and do not necessarily reflect those of the European Union or the European Climate, Infrastructure and Environment Executive Agency (CINEA). Neither the European Union nor the granting authority can be held responsible for them.

NW and JFD acknowledge support from the European Research Council Advanced Grant project ERA (Earth Resilience in the Anthropocene, ERC-2016-ADG-743080).

The manuscript was refined using AI-assisted tools (ChatGPT, OpenAI).

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
