# Peer review of "Quantifying Resilience in Non-Autonomous and Stochastic Earth System Dynamics with Application to Glacial-Interglacial Cycles"

_EGUsphere, 2025_

## Referee Comment (RC1)

**Review Report for "Quantifying Resilience in Non-Autonomous and Stochastic Earth System Dynamics with Application to Glacial–Interglacial Cycles" by Jakob Harteg et al.**

Reviewer: Takahito Mitsui

December 2025

**General Comment**

Recently, Earth system resilience has attracted increasing attention from the research community, driven by growing awareness of climatic tipping points. However, existing theories of Earth system resilience have mainly been developed for autonomous dynamical systems with fixed-point attractors, and therefore cannot be straightforwardly applied to non-autonomous phenomena such as glacial–interglacial cycles.

In this context, the authors develop and demonstrate a methodology for measuring *path-wise* Earth system resilience, focusing on late-Pleistocene ice-age cycles. They propose two metrics to characterize such path-wise resilience: the Reference Adherence Ratio (RAR), defined as the fraction of stochastic trajectories that remain within a narrow band around an unperturbed reference trajectory, and the return time, defined as the time required for a single perturbed trajectory to return to the reference path. Using a simple conceptual model of ice-age cycles, the authors demonstrate that these metrics can reveal differences in resilience across various glacial and interglacial epochs.

The reviewer appreciates the direction of this research. The manuscript is well structured and clearly written, and the limitations associated with model artefacts are explicitly stated. However, the advantages of the proposed metrics over the stability indicators used in previous studies are not sufficiently clear. In addition, the model equations appear to contain internal inconsistencies, and the practical use of the proposed resilience metrics is not entirely clear.

Overall, this reviewer considers that the manuscript has the potential to be published, but identifies several issues that should be clarified or addressed prior to publication.

**Major Issues**

1) **Inconsistency in the model equations.** When $\frac{dv}{dt} \geq 0$, Eq. (3) is given as

$$CO_2 = c_1 T - c_2 v + c_4.$$

Eliminating $T$ using Eq. (4),

$$T = d_1 v + d_2 \log\left(\frac{CO_2}{278}\right),$$

yields

$$CO_2 = (c_1 d_1 - c_2)v + c_1 d_2 \log\left(\frac{CO_2}{278}\right) + c_4.$$

This expression implies

$$v = \frac{CO_2 - c_1 d_2 \log\left(\frac{CO_2}{278}\right) - c_4}{c_1 d_1 - c_2},$$

which contradicts the dynamical equation for $v$ given in Eq. (1). A similar issue also arises for the case $\frac{dv}{dt} < 0$.

Given the excellent agreement between the model output and the ice volume reconstruction, the reviewer believes in the scientific value of the model. Nevertheless, such an inconsistency at the level of the governing equations should be clarified or resolved.

2) **Relation to previous studies on stability and resilience.** This reviewer views the work by Harteg et al. as valuable in that it explicitly places glacial–interglacial cycles within the modern framework of Earth system resilience theory. On the other hand, the resilience or (in)stability of glacial–interglacial trajectories has been discussed in several earlier studies.

Finite-time Lyapunov exponents have been used as indicators of transient instability in De Saedeleer et al. (2013), Mitsui and Aihara (2014), and Mitsui and Crucifix (2016). In these studies, the finite-time Lyapunov exponent increases prior to the temporal separation of unperturbed and perturbed trajectories (e.g., Fig. 8 in Mitsui and Aihara, 2014; Fig. 6 in Mitsui and Crucifix, 2016). This behavior appears slightly different from that of the RAR. For instance, the RAR remains nearly constant during trajectory separation near MIS 3 in Fig. 3 of the present manuscript. One might ask whether log(RAR) could serve as a more sensitive indicator.

In addition, so-called potential analysis has revealed temporal changes in attractors or basin structures across glacial–interglacial cycles (Livina et al. 2011). Dakos et al. (2008) discussed conventional early-warning signals prior to deglaciations, although these may not be applicable to strongly non-equilibrium systems. A clearer comparison with these existing approaches would strengthen the manuscript.

3) **Importance of the proposed resilience measures.** In previous studies, finite-time Lyapunov exponents have been used to explain the existence of periods during which the simulated ice-age trajectory is vulnerable to perturbations. Although this article emphasizes the necessity of path-wise resilience concepts, it remains unclear how the proposed RAR and the return time can be used in practice. For example, the reviewer wonders how these measures can be useful for assessing the safety of long-term storage and disposal of nuclear waste, which was one of the original motivations of the work by Talento and Ganopolski (2021), whose model is adopted in Section 2. If this question is beyond the scope of the present study, it can be safely ignored.

**Minor Issues**

Section 3.1.3: This study assumes a unique deterministic trajectory in the absence of perturbations. However, previous work has demonstrated the possible coexistence of multiple trajectories (e.g., De Saedeleer et al. 2013). Moreover, in more complex climate models (including CLIMBER-2), deterministic trajectories can be chaotic and may depend on initial conditions, although they may remain largely synchronized with astronomical forcing. In such cases, defining a reference trajectory $v_{\mathrm{ref}}(t)$ becomes non-trivial. This issue does not necessarily undermine the usefulness of the RAR; however, it would be helpful if the authors discussed how the proposed framework could be applied in such situations.

Line 121: "In the model, orbital forcing $f$ represents maximum summer insolation at 65°N." Conventionally, maximum summer insolation at 65°N refers to the mean daily insolation at the summer solstice, for which the relative contributions of obliquity and climatic precession are defined *a priori*. In Eq. (5), however, their relative weights are tuned. This distinction should be clarified.

Line 124: It would be helpful to describe how pre($t$) and obl($t$) are preprocessed. The raw amplitudes of climatic precession and obliquity (expressed in radians or degrees) differ substantially.

In the caption of Fig. 1, $f$ is the orbital forcing and $f'$ is its anomaly. However, $f$ appears to be anomaly in Eq. (5). Also mean($f$) and $\overline{f}$ coexist.

Line 139: "This change is not well explained by orbital forcing alone" in this model.

Line 148: The term "a steady state" is not appropriate for a non-autonomous system. A pullback attractor, or simply a trajectory after removal of transients, would be more accurate.

Line 173: $dW_t$ is not the Wiener process; $W_t$ denotes the Wiener process.

Figure 2: "Mean period" → "Period"?

**References**

1. De Saedeleer, B., Crucifix, M., & Wieczorek, S. (2013). Is the astronomical forcing a reliable and unique pacemaker for climate? A conceptual model study. *Climate Dynamics*, 40(1), 273–294.

2. Mitsui, T., & Aihara, K. (2014). Dynamics between order and chaos in conceptual models of glacial cycles. *Climate Dynamics*, 42(11), 3087–3099.

3. Mitsui, T., & Crucifix, M. (2016). Effects of additive noise on the stability of glacial cycles. In *Mathematical Paradigms of Climate Science* (pp. 93–113). Springer.

4. Livina, V. N., Kwasniok, F., Lohmann, G., Kantelhardt, J. W., & Lenton, T. M. (2011). Changing climate states and stability: from Pliocene to present. *Climate Dynamics*, 37(11), 2437–2453.

5. Dakos, V., Scheffer, M., Van Nes, E. H., Brovkin, V., Petoukhov, V., & Held, H. (2008). Slowing down as an early warning signal for abrupt climate change. *Proceedings of the National Academy of Sciences*, 105(38), 14308–14312.